# Case Report: Spontaneous Pneumomediastinum and Pneumothorax Complicating Severe Ketoacidosis—An Unexpected Presentation

**DOI:** 10.3390/reports8020095

**Published:** 2025-06-18

**Authors:** Alexandru Cristian Cindrea, Adina Maria Marza, Alexandra Maria Borita, Antonia Armega-Anghelescu, Ovidiu Alexandru Mederle

**Affiliations:** 1Doctoral School, Faculty of General Medicine, “Victor Babes” University of Medicine and Pharmacy Timisoara, 300041 Timisoara, Romania; alexandru.cindrea@umft.ro (A.C.C.); antonia.armega@umft.ro (A.A.-A.); 2Emergency Clinical Municipal Hospital, 300254 Timisoara, Romania; alexandraborita@gmail.com (A.M.B.); mederle.ovidiu@umft.ro (O.A.M.); 3Department of Surgery, Emergency Discipline, “Victor Babes” University of Medicine and Pharmacy, 300041 Timisoara, Romania

**Keywords:** interdisciplinary, diabetic ketoacidosis, mediastinal emphysema, pneumothorax, pancreatitis

## Abstract

**Background and Clinical Significance:** Diabetic ketoacidosis (DKA) is a serious and potentially life-threatening condition, often triggered by infections or undiagnosed diabetes. Spontaneous pneumomediastinum (SPM) and pneumothorax are rare but recognized complications of DKA, possibly due to alveolar rupture from increased respiratory effort or vomiting. Sometimes, acute pancreatitis (AP) may further complicate DKA, but the co-occurrence of these three conditions remains exceptionally rare. **Case Presentation:** We describe the case of a 60-year-old woman without a known history of diabetes who arrived at the emergency department with abdominal pain, fatigue, vomiting, and altered mental status. Initial laboratory findings showed metabolic acidosis, hyperglycemia, and elevated anion gap, consistent with DKA. Imaging revealed spontaneous pneumomediastinum and subsequently a left-sided pneumothorax, without evidence of trauma or esophageal rupture. Epigastric pain, along with elevated serum lipase and CT findings, also confirmed acute pancreatitis. Despite the complexity of her condition, the patient responded well to supportive treatment, including oxygen therapy, fluid resuscitation, insulin infusion, and antibiotics. She was discharged in good condition after 28 days, with a confirmed diagnosis of type 2 diabetes, without further complications. **Conclusions:** This case highlights an unusual combination of DKA complicated by spontaneous pneumomediastinum, pneumothorax and acute pancreatitis in a previously undiagnosed diabetic patient. Because prompt intervention can lead to favorable outcomes even in complex, multisystem cases, early recognition of atypical DKA complications is critical in order to avoid misdiagnosis.

## 1. Introduction and Clinical Significance

Diabetic ketoacidosis (DKA) is a life-threatening hyperglycemic emergency, characterized by the triad of hyperglycemia, increased blood or urine ketone concentration and metabolic acidosis [1,2]. This pathology accounts for 14% of all hospital admissions and 16% of all the deaths in patients with known diabetes [3]. Patients are at risk of developing DKA regardless of their diabetes mellitus (DM) type [4], even though it is more frequent in patients with type 1 DM [5].

Spontaneous pneumomediastinum (SPM) is a rare, usually benign, condition defined as the presence of free air within the mediastinum, in the absence of identifiable tracheobronchial trauma [6,7,8,9]. Pneumomediastinum has a reported prevalence of less than 1 out of 44,000 emergency department (ED) visits [8], being more frequent in young males [10]. Mortality associated with SPM is usually low and associated with the development of mediastinitis [11]. The association between SPM and DKA has been previously documented in the literature [12,13,14,15,16,17,18,19], with the first mention being made by Hamman in 1939 [19]. The largest study on DKA complicated with SPM was published by Zhang et al. [15], encompassing 79 patients. Out of those, five patients had associated epidural pneumatosis and three of them spontaneous pneumothorax (SP). The reported mortality was low, with the authors documenting it in only two cases. In another case series of 10 patients with SPM complicating DKA, described by Xu et al. [17], three patients did not have a medical history of DM and no deaths were reported.

Acute pancreatitis (AP) represents the inflammation of the pancreas, with a potentially life-threating evolution [20,21]. The reported age standardized incidence rate of AP varies significantly across different regions, between 1990 and 2019 varying from 8.7 to 82 cases per 100,000 individuals (higher rates in Eastern Europe) [22]. Mild cases have a low mortality rate of less than 1%, while severe acute pancreatitis can have mortality rates ranging from 10% to 30%, particularly in the presence of infected pancreatic necrosis or persistent organ failure [23]. In a 6-year multicentric study, Khan et al. [24] analyzed the outcomes of patients with DKA with coexistent AP. The reported incidence of AP was 9%. In this cohort, the association of DKA and AP significantly increased the median length of hospital admissions (2.4 vs. 4.8 days) and the rate of ICU admission (23.1% vs. 39.3%), with no remarkable effect on mortality, readmission and DKA recurrence.

This paper presents a rare and complex case of inaugural diabetic ketoacidosis complicated by spontaneous pneumomediastinum, spontaneous pneumothorax, and acute pancreatitis. To our knowledge, the simultaneous occurrence of all three complications in the context of DKA has not been previously reported. This case aims to raise awareness among clinicians about this unusual triad, emphasizing the importance of early recognition and appropriate management to avoid misdiagnosis and unnecessary interventions.

## 2. Case Presentation

A 60-year-old Caucasian woman with no documented pre-existing medical conditions, body mass index 25.1 kg/m^2^, presented to the emergency department (ED) with fatigue, nausea, a single episode of vomiting, loss of appetite, unintentional weight loss, upper-level abdominal pain and sopor. Clinical examination revealed warm peripheries, confusion and peri-umbilical and epigastric pain on superficial palpation. Vital signs on admission were blood pressure of 140/95 mmHg, T = 36.1 °C, heart rate of 151 beats per min and respiratory rate of 28 breaths per min.

After the patient’s evaluation (following the Airway–Breathing–Circulation–Disability–Exposure assessment), a localized area of subcutaneous crepitus was noted in the lateral cervical region, without a history or clinical signs of trauma. Chest expansion was symmetrical bilaterally, pulmonary percussion demonstrated normal resonance, and auscultation revealed normal breathing sounds without any rales. Peripheral oxygen saturation on ambient condition was 93%. These findings raised concerns about a possible small SP, and a thoracic computed tomography (CT) scan was planned.

Two large-bore intravenous lines were placed, blood specimens were collected and fluid resuscitation was initiated. The abdomen exhibited respiratory mobility, with no postoperative scars, and was tender on deep palpation. The Glasgow coma scale was 13.

The initial investigations were the acid/base status, electrolyte and metabolites, which revealed the following findings: pH 7.06, pressure of O_2_ 18.4 mmHg, pressure of CO_2_ 28.2, base excess −19.5 mmol/L, bicarbonate (HCO^3−^) 8.4 mmol/L, anion gap 33.1 mmol/L, blood glucose 559 mg/dL, lactate 2.44 mmol/L, sodium (Na^+^) 140.7 mmol/L, potassium (K^+^) 3.76 mmol/L, calcium (Ca^2+^) 1.29 mmol/L, chloride (Cl^−^) 103 mmol/L. Based on these results, a diagnosis of diabetic ketoacidosis was considered the most probable. A urine sample was collected after urinary catheterization was performed, and treatment for the suspected condition was initiated promptly, in accordance with the DKA guidelines. Fluid resuscitation was initiated at a rate of 15 mL/kg/h. Subsequently, fluid management was adjusted to compensate for ongoing losses. A twelve-lead electrocardiogram revealed sinus tachycardia with a ventricular rate of approximately 166 beats per minute, without evidence of ischemic changes.

The initial blood analyses (taken in the ED) and the following values taken on the ward are depicted in Table 1. The urine exam revealed glucosuria (>1000 mg/mL) and ketonuria (>160 mg/dL). Given the high values of lipase (increased by almost 10 times) accompanied by epigastric pain, acute pancreatitis was also suspected.

The chest CT scan performed in the emergency department (Figure 1) revealed the presence of pneumomediastinum. CT scans of the abdomen and pelvis were also performed, giving the high lipase and epigastric pain, revealing findings consistent with acute pancreatitis (with peripancreatic fluid collection, consistent with moderate acute pancreatitis per the modified CT severity index), as well as a solitary gallbladder stone measuring approximately 20 mm in diameter, without intra or extrahepatic biliary ducts enlargement (Figure 2).

Given the patient’s history of vomiting, and consulting with the thoracic surgeon, a second CT scan with oral contrast was performed approximately 3 h later, after initial stabilization of the patient, to rule out an esophageal rupture. The subsequent imaging revealed the progression of the pneumomediastinum, with increased extension into the cervical region, including the peritracheoesophageal, perithyroidal and retropharyngeal spaces (Figure 3), without evidence of esophageal rupture (Figure 4). Oxygen therapy and empiric antibiotic therapy were initiated in the ED as initial treatments of this condition.

The patient was referred to diabetology in order to be admitted to hospital. Although the patient presented a good clinical and biological evolution, her condition deteriorated after about 12 h; therefore, another chest CT scan was performed, revealing extension of the pneumomediastinum in comparison to the prior scan and the presence of an additional left pneumothorax measuring approximately 7 mm (Figure 5).

The evolution was favorable, with the general state of the patient improving during the admission. The patient became fully cooperative relatively early during the hospital stay. The subcutaneous emphysema resolved approximately three days after admission. Supplementary investigations performed consist of serial blood glucose measurements (with a 24 h mean glycemia of 117 mg/dL), HbA1C (10.4%), C peptide (2.650 ng/mL, normal range 0.810–3.850 ng/mL), triglycerides (163 mg/dL) and free triiodothyronine (3.13 pmol/mL).

A repeat chest and abdominal CT scan was performed seven days after admission. It revealed resolution of the pneumomediastinum, with only small millimetric air inclusions remaining at the level of the thymic lodge. The appearance of the pancreas remained unchanged at this time.

The patient was discharged after 18 days of hospitalization, after complete resolution of the pneumomediastinum, presenting no residual symptoms. A pancreatic pseudocyst, which developed during the hospital stay, diagnosed via abdominal CT scan with intravenous contrast, was the only remaining finding. Discharge recommendations included oral antidiabetic therapy and adherence to a dietary regimen.

## 3. Discussion

The current paper presented the case of a 60-year-old woman, with no known medical history, who presented in the ED with classic signs of severe DKA. Additionally, the patient presented SPM, SP and AP associated with the ketoacidosis. The symptomatology developed about 2 weeks prior to the ED presentation. Furthermore, the patient presented HbA1C levels of 10.4% suggesting a prolonged evolution of the disease. Type 2 DM was confirmed using the C peptide levels.

### 3.1. Inaugural DKA in Adult Patients

DKA represents about 38% of the total number of hospital admissions for hyperglycemic crisis in diabetic patients, with a peak incidence in adults aged 18–44 years with type 1 DM [1]. Infections, particularly pneumonias and urinary tract infections, are widely recognized as the most common precipitating factors for DKA [25,26,27]. Inaugural DKA represents the development of DKA in a patient with no previous diagnosis of DM. In a recently published study, Sall et. al. reported a prevalence of inaugural diabetic ketoacidosis of 17.1% in adult patients [5], while other studies reported rates of up to 20% [25]. In this case report, the patient was a 60-year-old-woman, with classic signs of diabetic ketoacidosis. Even though she was not previously diagnosed with DM, the patient presented HbA1C levels of 10.4% suggesting a prolonged evolution of the disease.

DKA is the result of a reduced or absent function of pancreatic beta cells to secrete insulin, along with an exaggerated response of the liver to compensate for this energy deficit. As a consequence, fatty acid metabolism is increased, leading to the accumulation of ketoacids (acetoacetate and β-hydroxybutyrate) [28].

Severity of the disease is defined by the American Diabetes Association taking into account the degree of acidosis and level of consciousness, the severe diseases being characterized by a pH below 7.0, bicarbonate levels below 10, a positive urine or serum nitroprusside test, an anion gap above 12 and presence of either stupor or coma [29]. The patient from the current case report respects all the criteria, except the pH values (which were slightly higher, 7.04).

Ketosis-prone diabetes (sometimes referred to as Flatbush disease) is a syndrome characterized by inaugural diabetic ketoacidosis in patients without typical characteristics of autoimmune type 1 DM [30]. It is highly prevalent in African and Hispanic people [31,32] and accounts for about one third of inaugural DKA or new onset ketosis in adults [33]. The underlying mechanism is believed to be the reversible, transient, destruction of the alpha and beta pancreatic cells [31]. Most patients with ketosis-prone diabetes become insulin independent and their pathology can be managed utilizing diet alone or in association with oral antidiabetics [34]. Older patients with obesity and a family history of type 2 DM, who present with DKA, must raise suspicion of ketosis-prone diabetes [31]. The pathology puts patients at risk of recurrent episodes of DKA, therefore supplemental attention is needed to diagnose and observe atypical forms of DM. Considering that hyperglycemic event-related visits in the ED are in a permanent growth, supplemental attention is required to prevent these complications. Benoit et al. observed an annual percentage change of 13.5% of adult patients presenting with DKA in the ED [35]. Proper diagnosis of this pathology is required in order to ensure adequate follow-up, avoiding unnecessary acute hyperglycemic crisis in this patient population.

### 3.2. DKA-Associated Spontaneous Pneumomediastinum

As previously mentioned, SPM is a rare condition and represents the presence of air in the mediastinum, with or without subcutaneous emphysema, in the absence of trauma. It is an uncommon but increasingly recognized complication of DKA. In this setting, the combination of severe metabolic acidosis, forceful vomiting, and Kussmaul breathing can lead to alveolar rupture, allowing air to dissect along bronchovascular bundles into the mediastinum, phenomenon known as the Macklin Effect. This process represents an atypical manifestation in patients presenting with DKA, underscoring the importance of maintaining a high index of suspicion in the appropriate clinical context [12]. Advanced imaging modalities, particularly computed tomography, are pivotal in confirming the diagnosis by revealing air tracking along the bronchovascular sheaths, thus differentiating it from other causes of mediastinal air accumulation [14,17].

Despite its generally benign and self-resolving course, the presence of pneumomediastinum in DKA is clinically significant because it may mimic more severe conditions such as esophageal rupture, which carries a high mortality rate if not promptly identified and managed [13,18]. The most frequent symptoms are chest pain and cough, followed by dyspnea, dysphagia, neck and throat pain or dysphonia [36].

In our case, esophageal rupture was excluded promptly using oral contrast CT. Consequently, the pneumomediastinum in this case could have been caused by the increased work of breathing, which might have been sustained for a longer period of time. Patients with DKA develop tachypnea as a compensatory mechanism for the metabolic acidosis. As acidosis progresses, patients develop hyperpnea (increased tidal volume), which eventually will lead to a deep, agonal breathing, Kussmaul’s respiration and acute respiratory distress syndrome [37]. As a consequence, this patient could have developed self-inflicted lung injury (P-SILI). P-SILI refers to lung damage that occurs due to a patient’s own excessive respiratory efforts, such as those seen in conditions that cause tachypnea. Increased respiratory drive can lead to substantial negative pressures within the lungs, resulting in various forms of injury [38,39,40,41]. Mechanisms underlying P-SILI include, but are not limited to overdistension of alveoli, Pendelluft Phenomenon and increased trans vascular pressure [39]. Sklienka et al. [41] mention that hyperventilation lasting for a prolonged period led to hypoxemia, decrease of lung compliance and structural features characteristic to the acute respiratory distress syndrome.

In this case, both the Maklin Effect and P-SILI are possible mechanisms for developing SPM. Similar underlying mechanisms could be implicated in the development of several other pneumo-complications reported in the literature to be associated with DKA: SP, pneumopericardium [15,18,36,42,43,44], pneumoperitoneum [45] and epidural pneumatosis (pneumorrhachis) [12,44].

Given that the pathogenesis of SPM secondary to DKA is complex and most patients with DKA frequently present one or more of tachypnea, Kussmaul’s breathing or vomiting, some authors proposed that many cases of this association could have remained undiagnosed [46]. Additionally, SPM complicating DKA is typically a self-limiting condition, and esophageal rupture is rarely implicated in its pathogenesis. In a review of 79 cases by Zhang et al. [15], only one patient was found to have a tear in the cardia, diagnosed via gastroscopy. Similarly, the study conducted by Pauw et al. [47], which analyzed 56 published cases, reported no instances of esophageal rupture or similar complications.

### 3.3. Acute Pancreatitis and DKA

Coexisting acute pancreatitis and DKA may be present in 10% to 15% of the cases [48]. Acute pancreatitis has been reported as both a cause and a complication of DKA [49]. The increased degree of lipolysis in the adipose tissue promoted by DKA causes severe hypertriglyceridemia, which is known to precipitate acute pancreatitis usually when surpassing 1000 mg/dL [50]. Acidosis (which promotes trypsinogen activations) and hyperviscosity are other incriminated factors in developing acute pancreatitis [51]. Tollard et al. [52] presented the case of a 32-year-old obese female patient who developed acute pancreatitis as a complication of DKA probably precipitated by SARS-CoV-2 infection. In this particular case, the underlying mechanism was hypothesized to have been the pancreatic cellular damage caused by the virus, as other causes of acute pancreatitis were excluded.

Alternatively, stress hyperglycemia in acute pancreatitis triggers a cascade of neuroendocrine and inflammatory responses that sharply elevate glucose levels and can precipitate DKA in insulin-intolerant individuals [50]. In this case, the only possible cause of acute pancreatitis identified was the presence of a gallbladder stone. Yet, the risk of developing acute pancreatitis is increased by small stones and multiple stones [53,54], rather than a large solitary stone.

It is worth mentioning that nonspecific elevations of amylase and lipase may occur in the context of DKA, maybe due to the reduced glomerular filtration rate present in hyperglycemic crises [48].

Walled-off pancreatic necrosis (WON) is a well-circumscribed encapsulated collection of fluid and necrotic debris that typically occurs more than four weeks after an episode of acute necrotizing pancreatitis, and while it may remain asymptomatic, about one-third of cases eventually require interventional treatment due to complications such as infection or rupture [55]. This complication should be suspected in patients with a history of acute necrotizing pancreatitis who, after 4 weeks, present with persistent or new-onset abdominal pain, fever, sepsis, or imaging evidence of a mature, encapsulated necrotic fluid collection [55,56,57]. Although in this particular case the prolonged hospitalization and the presence of pancreatic fluid collections could have raised clinical suspicion for WON, the diagnosis was initially excluded based on abdominal CT with intravenous contrast. Nevertheless, the persistence of a pancreatic pseudocyst remains concerning and warrants further close monitoring and follow-up imaging to evaluate for potential evolution into WON or other complications.

## 4. Conclusions

Diabetic ketoacidosis as an inaugural presentation of type 2 diabetes mellitus is relatively uncommon and often challenging to manage. However, this case proved to be significantly more complex when imaging diagnostics revealed three additional, potentially fatal, diagnoses: SPM, SP and AP. This case report aims to highlight the necessity of promptly corelating clinical, biological and imaging data. Timely diagnosis and targeted interventions can significantly improve outcomes, as demonstrated by the favorable evolution of this case. However, given the limited reports on this topic in older patients, the long-term evolution of such a complex pathology remains uncertain, underlining the need for further clinical observation and research.

## Figures and Tables

**Figure 1 reports-08-00095-f001:**
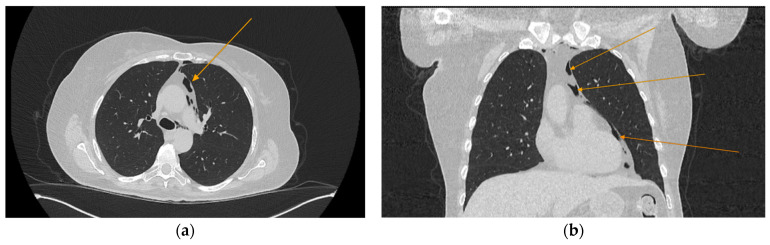
Initial chest CT scan, demonstrating pneumomediastinum (arrows): (**a**) axial view showing free air in the mediastinal space; (**b**) coronal view revealing air tracking along several mediastinal structures.

**Figure 2 reports-08-00095-f002:**
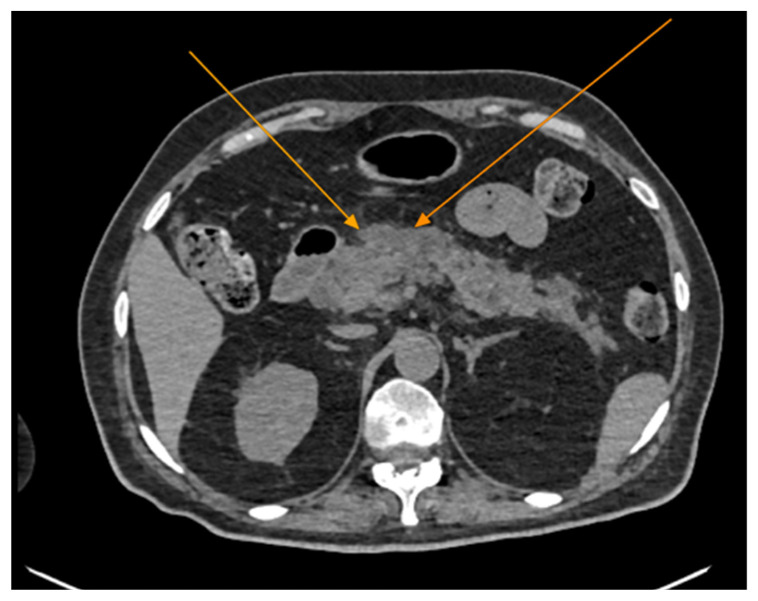
Abdominal CT scan demonstrating features of acute pancreatitis (arrows), axial view. Note the presence of peripancreatic fluid collections.

**Figure 3 reports-08-00095-f003:**
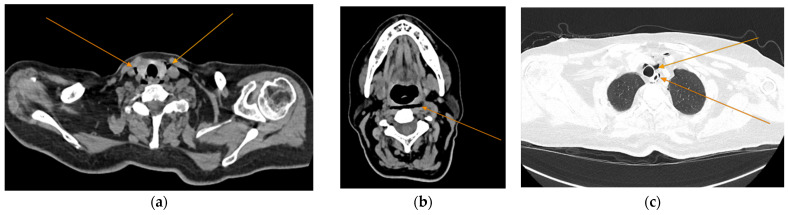
Repeated chest and neck CT scans showcasing extension of the pneumomediastinum (arrows), axial views: (**a**) air tracking into the cervical soft tissues and prevertebral space; (**b**) air around airway structures; (**c**) pneumomediastinum surrounding the trachea.

**Figure 4 reports-08-00095-f004:**
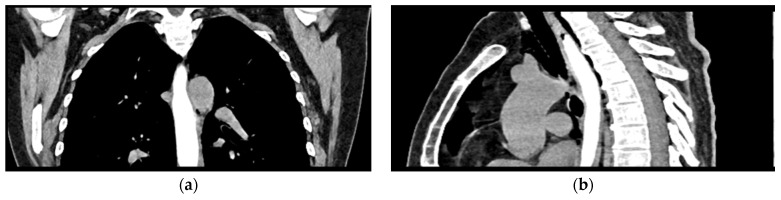
Oral contrast chest CT scan with intact esophageal contour and no extravasation of the contrast substance: (**a**) coronal view; (**b**) sagittal view.

**Figure 5 reports-08-00095-f005:**
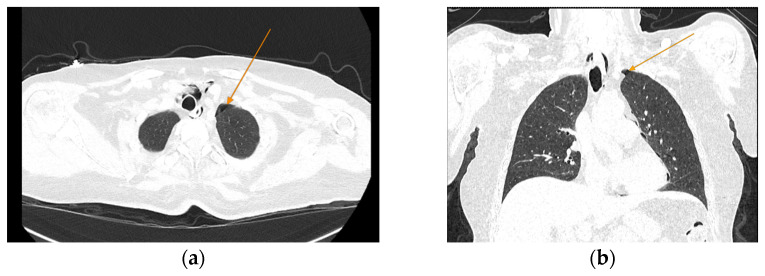
Chest CT showing the presence of pneumothorax (arrows): (**a**) axial view; (**b**) coronal view.

**Table 1 reports-08-00095-t001:** Sequential Laboratory Analyses of the Patient (Admission to Discharge).

Bloodwork	Reference Range and Measure Unit	Initial	7 days	Discharge
White blood cells	4–10 × 10^3^/µL	29.55	23.35	7.26
Neutrophiles	2–7 × 10^3^/µL	24.19	19.34	4.74
Hemoglobin	12–15 g/dL	16.09	14.25	12.32
Platelets	150–410 × 10^3^/µL	473.2	230.5	454
Potassium	3.5–5.1 mmol/L	3.5	2.3	4.3
Sodium	136–145 mmol/L	137	133	135
Creatinine	0.55–1.02 mg/dL	1.58	0.6	0.6
Urea	15–39 mg/dL	90	27	6
Glycemia	74–106 mg/dL	569	390	110
Lipase	16–77 U/L	687	421	213
ALT	14–59 U/L	47	25	20
AST	15–37 U/L	15	18	18
Total Bilirubin	0.2–1.00 mg/dL	0.66	0.9	0.5
C reactive protein	0–5 mg/L	34.4	73	34.3
Procalcitonin	0.1–0.25 ng/mL	0.14	-	-

Abbreviations: ALT—alanine aminotransferase; AST—aspartate aminotransferase.

## Data Availability

The original contributions presented in this study are included in the article. Further inquiries can be directed to the corresponding author.

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
