# Peer review of "Case Report: Spontaneous Pneumomediastinum and Pneumothorax Complicating Severe Ketoacidosis—An Unexpected Presentation"

_reports, 2025, doi:10.3390/reports8020095_

Round 1
Reviewer 1 Report
Comments and Suggestions for Authors
This case report preset a Spontaneous Pneumomediastinum and Pneumothorax Compli-2 cating Severe Diabetic Ketoacidosis: An Unexpected Presenta-3 tion 4 and tried to explain the rare association between two entities.
This is a rare case of this association, but in the same time the authors present other cases in references list. Sometimes, acute pancreatitis (AP) may fur-18 ther complicate DKA, but the co-occurrence of these three conditions remains exceptionally rare. Other co-occurrence of DKA and other 3-4 conditions remains rare, but the association could be in direction of an initial AP, initial SPM and finally DKA. It is rare DKA and next step AP or SPM (or SPM and AP).
The authors should try to explain some pathophysiology of these three associations.
This report present the Mortality associated with SPM is usually low and associated with the development of 48 mediastinitis, too.
This report said too: Acute pancreatitis (AP) represents the inflammation of the pancreas, with a poten-57 tially life-threating evolution.
In the same time the authors said “This paper presents a rare and complex case of inaugural diabetic ketoacidosis com-68 plicated by spontaneous pneumomediastinum, spontaneous pneumothorax, and acute 69 pancreatitis.”
Please, try to present the real evolution of this case.
We know that DKA is present at beginning?
Clinical evolution is not presented. Only lab evolution is presented.
Imagine reports at beginning, at 7 days and at discharge is not presented.
For a case report, the methodology is quite adequate and pretty well expressed, but a time evolution should be here.
For a report, we don’t have conclusions - how many liquids are used? A DKA is a common presentation of a TYPE 1 DM. But association could be present. SPM without DKA is present when?
References are appropriate.
The references are carefully chosen, from impact journals. They are updated, from the last 10-12 years, in the area of diabetes mellitus first stages.
Tables are very easy to be read them, to be understanded. Presented data are simple, a real evolution in time, it is very simple and detailed at the same time, supporting the conclusion of the report.
Author Response
see below

Reviewer 2 Report
Comments and Suggestions for Authors
The authors presented unusual case of inaugural DKA associated with spontaneous pneumothorax, pneumonomediastinum and acute pancreatitis. The case is important for everyday clinical practice.
Minor issues:
- Who is C.G.W? This name is not listed in the authors row
- 77 please, change obtundation into somnolence or sopor
- 86 change into normal breathing sound
- 86/87 change into ambient condition oxygen saturation
- change into chloride
- 97/98 delete was performed
- 101 change blood work into blood analyses
- 106 rename the title
- please, check the numbering of figures and their explanations
After corrections, the paper is suitable.
Comments on the Quality of English Language
English must be refined.
Author Response
See below

Reviewer 3 Report
Comments and Suggestions for Authors
It is a very interesting rare case of a 60-year-old woman with diabetic ketoacidosis, spontaneous pneumomediastinum and pneumothorax. The woman had not a known history of diabetes. Her C-peptide levels were high and her BMI was 25.1 kg/m2. The case is presented in a well-structured manner. The figures and the table are appropriate. The cited references are recent publications and relevant.
I only have a question: did you measure autoantibodies for type 1 diabetes? It would be interesting to know if the woman had positive autoantibodies.
Author Response
See below

Round 2
Reviewer 1 Report
Comments and Suggestions for Authors
The authors updated all recommended suggestions, improving the quality of manuscript.
In the same time, they explain some decision to maintain the part of the manuscript as it is.